# Oxidative Stress: Concept and Some Practical Aspects

**DOI:** 10.3390/antiox9090852

**Published:** 2020-09-10

**Authors:** Helmut Sies

**Affiliations:** 1Institute of Biochemistry and Molecular Biology I, Heinrich-Heine-University Düsseldorf, University Street 1, Bldg 22.04, D-40225 Düsseldorf, Germany; sies@hhu.de; 2Leibniz Research Institute for Environmental Medicine, Heinrich-Heine-University Düsseldorf, D-40225 Düsseldorf, Germany

**Keywords:** oxidative stress, antioxidants, biomarkers

## Abstract

Oxidative stress is defined as “an imbalance between oxidants and antioxidants in favor of the oxidants, leading to a disruption of redox signaling and control and/or molecular damage”. This Commentary presents basic features of this global concept which has attracted interest in biology and medicine. The term “antioxidants” in cellular defense against oxidants predominantly includes antioxidant enzymes with their substrates and coenzymes. Exogenous low-molecular-mass compounds also have a role, but this is more limited. Multiple biomarkers of damage due to oxidative stress have been identified for different molecular classes (protein, lipid, carbohydrate, and DNA), and the current state of practical aspects in health and disease is delineated.

## 1. Introduction

“Oxidative stress” is a global concept in redox biology and medicine. Since its introduction in 1985 [1], it has attracted widespread interest and also some critical comments [2], and it is covered in detail in a textbook [3] (pp. 199–283). The literature on oxidative stress is voluminous, and comprehensive coverage is not the subject of this Commentary; for example, the active fields of oxidative stress research in microbes and in plants are not covered, but the general principles apply to these fields as well. Rather, an attempt will be made to address the current state of knowledge in terms of practical aspects. Due to the multitude of oxidant and antioxidant processes occurring simultaneously in normal and pathophysiological conditions in different cells and organs, there have been numerous endeavors of translational application in a wide range of health and disease situations. Some of these have been based on oversimplifications. I will address this issue with some examples, focusing on recent literature.

## 2. Oxidative Stress

A brief survey of the literature in the Web of Science gives over 450,000 hits for the term “oxidative stress”, with about 40,000 new additions in 2020. Unsurprisingly, a comprehensive view on the current development of this term would be challenging.

### 2.1. The Concept of Oxidative Stress

The global concept of “Oxidative Stress” is defined as “an imbalance between oxidants and antioxidants in favor of the oxidants, leading to a disruption of redox signaling and control and/or molecular damage” [4,5]. It has developed from its initial formulation in 1985 [1] to incorporate new knowledge on the role of redox signaling [2]. The basic idea is that, in the open metabolic system, a steady-state redox balance is maintained at a given setpoint, which provides a basal redox tone, and that a deviation from the steady-state redox balance is considered a stress, initiating a stress response. Implicit in the definition of oxidative stress is (i) that a deviation to the opposite side of the balance is “reductive stress”, and (ii) that there are physiological deviations, “oxidative eustress”, and supraphysiological deviations, “oxidative distress” [5]. Oxidative eustress is an essential part of redox control and physiological redox signaling [6,7] (Figure 1). This concept overlaps with that of redox homeostasis as the “golden mean” [8]. Current knowledge on these topics has been presented in the book, “Oxidative stress: eustress and distress” [9].

### 2.2. Oxidative Stress: A Slice of the Pie

The molecular mechanisms of metabolic regulation include a multitude of chemical means. Structure and functions are implemented by orchestrated chemical modification of proteins, lipids, carbohydrates, and, not-the-least, nucleic acids. Redox reactions contribute to regulation, and it is noteworthy that there is crosstalk between different modes of regulation, e.g., between redox modifications and phosphorylation/dephosphorylation of proteins. Chemically reactive molecules of low-molecular-mass (“reactive species”) have been extensively investigated in their role in regulation. Recent accounts were given for reactive oxygen species (ROS) [7], reactive nitrogen species (RNS) [10,11], reactive sulfur species (RSS) [12], reactive electrophile species (RES) [13], and reactive halogen species (RHS) [14]. Clearly, multiple interactions constitute checks and balances in redox regulation [15].

## 3. Antioxidants

On the other side of the redox balance, the defense against damaging levels of oxidants consists of several types of antioxidant enzymes in conjunction with their back-up systems, as well as of low-molecular-mass antioxidants, forming an antioxidant network [16]. The complement of antioxidant enzymes is subject to regulation by redox master switches as part of the oxidative stress response (see Ref. [5] for a review). While these two preceding sentences appear to be uncontested, a widespread misconception about “antioxidants” and “antioxidant capacity” deserves a comment [17,18,19]. In lay language, the term antioxidant seems to be confined solely to exogenous low-molecular-mass compounds, neglecting the much more important contribution of antioxidant enzymes (with their substrates and coenzymes) to cellular defense against oxidants. This misconception also pervades in the scientific community, best exemplified by the term “antioxidant capacity”, or even “total antioxidant capacity (TAC)”. The misconception can be traced back to the work of excellent organic chemists, who investigated the capability of isolated human blood plasma to trap artificially generated peroxyl radicals [20]. This was the origin of the term “total antioxidant capacity” which, while true for isolated plasma in vitro, unfortunately is a misnomer as it was subsequently applied to the in vivo situation by many investigators. Such tests (TRAP, FRAP, ORAC, ABTS) do not assay any antioxidant enzyme capacity, and a more appropriate name instead of TAC would have been NEAC, non-enzymatic antioxidant capacity [21].

In their textbook, Halliwell and Gutteridge devoted a chapter to antioxidants synthesized in vivo [3] (pp. 77–152) and a separate one on antioxidants from the diet [3] (pp. 153–198), emphasizing the points made above: “Antioxidant is a term widely used but surprisingly difficult to define clearly”, and the simplified definition is “any substance that delays, prevents or removes oxidative damage to a target molecule” [3] (p. 77). Hopefully, the distinction between endogenous and exogenous antioxidants will guide more prudent research in the future.

## 4. Practical Aspects

Against this background, it is obvious that oxidative stress will have different intensities [22] and manifestations in different cell types and organs, as discussed previously [5,7,23]. The set of principles which defines the spatiotemporal positioning of the biological redox systems has been described as the “redox code” [24]. Redox patterns across cells and within subcellular spaces resemble a dynamic “landscape” rather than being flat [7,15,24].

### 4.1. Biomarkers

In accordance with the variety of compounds and pathways, numerous biomarkers of oxidative stress have been identified [25]. Frijhoff et al. [26] examined the relevance of specific biomarkers of oxidative stress in various diseases, as depicted in a cluster analysis (Figure 2).

Obviously, measuring a single biomarker is not clinically useful [27]. The epistemological challenges of using such biomarkers have been discussed [28]. Ghezzi [29] proposed a classification of biomarkers of oxidative stress (Table 1), which may help guide researchers in the field. A specific term worth pursuing further is “actionability of oxidative stress biomarkers”, i.e., whether a given biomarker is causally involved in the disease or not [29]. Advances in technology provide further opportunities for biomarkers, e.g., in redox phospholipidomics [30], proteomics [31], amino acids [32], protein lipoxidation [33], or metabolomics [34]. Gender- and age-dependencies of different oxidative stress biomarkers have been noted [35].

### 4.2. Clinical Aspects

There is a wide literature of oxidative stress manifestation in clinical settings, as presented in pertinent reviews [5,7,23,36,37,38,39]. In-depth discussion of these aspects is beyond the scope of this Commentary (see [40]). A promising emerging area for research relates to mind-body medicine [41], the relationship between psychological stress and oxidative stress [42,43]. Our recent coverage on oxidative stress manifestation in clinical settings [7] is on the following topics: central nervous system; immune system, inflammation and wound repair; cardiovascular system; skeletal muscle; insulin sensitivity and pathogenesis of diabetes; ROS in aging and lifespan regulation; cancer; and prospects for redox medicine. Redox systems analysis of antioxidant networks will help to further understand the interplay of nutrition and oxidative stress [44].

## 5. Conclusions

Understanding the mechanisms of maintenance of the redox balance has progressed in recent time, opening opportunities for redox-based strategies in practical applications in medicine (“redox medicine”). Due to the complexity of interlinked stress response systems, future research will focus on redox systems biology.

## Figures and Tables

**Figure 1 antioxidants-09-00852-f001:**
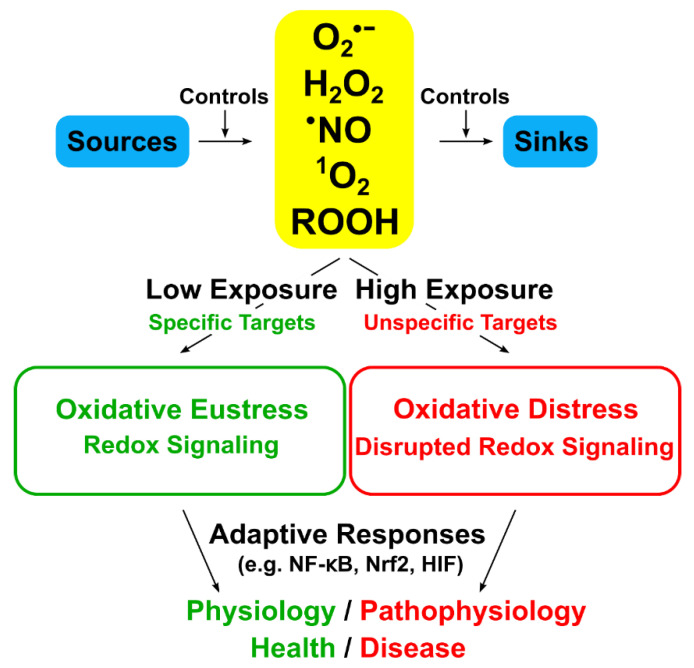
**Oxidative stress and its relationship to redox signaling**. Physiological (low) oxidant exposure addresses specific (highly-reactive) targets, whereas supraphysiological (high) exposure addresses unspecific targets. Adaptive responses counteract. Further important oxidants are generated in secondary reactions; e.g., ONOOH, from O_2_^·−^ and ^·^NO, or HOCl, from H_2_O_2_ and Cl^−^. Modified from Ref. [6]. Creative Commons License.

**Figure 2 antioxidants-09-00852-f002:**
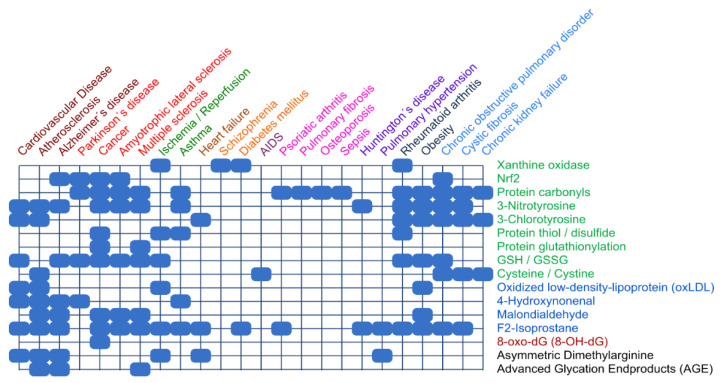
**Clinical relevance of biomarkers of oxidative stress.** Color coding on right: Protein (green), lipid (blue), and DNA (red) biomarkers are analyzed for various diseases. Color coding on top: results from cluster analysis. Data compiled from Frijhoff et al. [26], presented as Figure 11 in Ref. [5], with permission.

**Table 1 antioxidants-09-00852-t001:** Classification of biomarkers related to oxidative stress. Compiled from Ghezzi [29].

Type	Biomarker
Type 0	Direct measurement of specific ROS molecule (e.g., H_2_O_2_)
Type 1	Protein carbonyls; MDA, HNE, isoprostanes, oxLDL; 8-oxo-dG
Type 2	HOCl; uric acid, allantoin
Type 3	SOD, CAT, GPX, PON1, NOX, XO, DUOX, Vit. E, Vit. C, bilirubin; [note: caveat on “TAC” Refs. [18,19]
Type 4	Genetic factors and mutations

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
