# Peer review of "Oxidative Stress: Concept and Some Practical Aspects"

_antioxidants, 2020, doi:10.3390/antiox9090852_

Round 1
Reviewer 1 Report
This is a commentary paper about oxidative stress written by one of the most prominent scientists in this field. Oxidative stress is a hugely important topic which impinges on cellular function and therefore on a wide range of biological and medical sciences. There was an odd grammar mistake but nothing that can’t be solved in type setting/proof reading.
My only criticism is that this comes across as quite animal-centric, whereas in the abstract it says “…interest in biology and medicine.” Is there space for a short paragraph on non-animal systems?
In summary, it was an interesting read and I am sure will be useful for many people using this journal.
Author Response
Thank you for the comment.
I searched for the "odd grammar mistake", I could find only the missing "the" on line 25, which was pointed out by Reviewer 2.
The non-animal systems were mentioned in the sentence on line 23-26, an extra paragraph was not added for plant and microbe research.
Reviewer 2 Report
This is an interesting commentary that is likely to promote thought in the majority of readers. It is a worthwhile contribution.
A few minor points for consideration by the author:
1) It may be better to use 'low-molecular-mass' than small-molecular-weight as low is the opposite of 'high' and mass is the preferred IUPAC term.
2) Line 25: ... is not the subject
3) Line 68: 'complement' may be better than 'pattern'
4) Figure 1: whilst I understand the authors wish not to have en exhaustive list of species in the figure, it is perhaps strange not to include ONOOH and HOCl as these are both widely examined and important biological oxidants
Author Response
Thank you for the comments.
(1) low-molecular-mass is now replacing small-molecular weight
(2) "the" was added
(3) "complement" now replaces "pattern"
(4) ONOOH and HOCl are now mentioned on line 53/54 as further important oxidants.
Reviewer 3 Report
I am much honored to read your review.
This review clearly explains the concept of good oxidative stress and bad oxidative stress when thinking about oxidative stress.
This paper also emphasizes that it is difficult to evaluate only a single biomarker for understanding oxidative stress-related diseases.
In addition, this review details the biomarkers involved in each disease.
I think that this review can provide with the useful information to the basic researcher or the clinical expert about the treatment for oxidative stress-related diseases.
Author Response
Thank you for the kind comments.